# Attitude and Acceptance towards COVID-19 Booster Doses among Literacy Advantaged Population in Pakistan: A Cross-Sectional Study

**DOI:** 10.3390/vaccines11071238

**Published:** 2023-07-14

**Authors:** Mehmood Ahmad, Adeel Sattar, Sadaf Aroosa, Arfa Majeed, Muhammad Adil Rasheed, Waqas Ahmad, Asif Iqbal, Muhammad Ovais Omer, Bilal Mahmood Beg, Rana Muhammad Zahid Mushtaq

**Affiliations:** 1Department of Pharmacology and Toxicology, University of Veterinary and Animal Sciences, Lahore 54000, Pakistan; 2Department of Pharmacology, Riphah International University, Lahore 54000, Pakistan; 3Department of Pathology, University of Veterinary and Animal Sciences, Lahore 54000, Pakistan; 4Department of Parasitology, Riphah International University, Lahore 54000, Pakistan; 5Division of Infection Medicine, College of Medicine, The University of Edinburgh, Edinburgh EH16 4SB, UK

**Keywords:** attitude, COVID-19, vaccine, booster doses, Pakistan

## Abstract

Severe acute respiratory syndrome coronavirus-2 (SARS-CoV-2) has affected billions of lives and is expected to impose a significant burden on the economy worldwide. Vaccination is the only way to prevent the infection. However, convincing people to get themselves vaccinated is challenging in developing countries such as Pakistan. Therefore, a cross-sectional questionnaire-based study was conducted (*n* = 982 participants) all over Pakistan to evaluate the perception, knowledge, attitude, and acceptance of the general public towards the SARS-CoV-2 vaccine, in general, and a booster dose of SARS-CoV-2, in particular. The highest number of participants were from the province of Punjab (84.5%), followed by Islamabad (3.8%), Sindh (3.7%), Khyber Pakhtunkhwa (2.7%), Baluchistan (2.6%), Gilgit Baltistan (1.4%), and Azad Jammu and Kashmir (1.4%). A total of 915 participants were vaccinated against COVID-19, out of which 62.2% received one booster dose, followed by double booster doses (25.5%) and single vaccine shots (12.3%). The highest number of vaccinated participants were from Punjab (85.8%), followed by Islamabad (3.9%), Sindh (2.8%); Khyber Pakhtunkhwa (2.6%); Baluchistan (2.3%); Gilgit-Baltistan (1.3%); and Azad, Jammu, and Kashmir (1.2%). Among the vaccinated individuals, 71.4% were unemployed, 27.4% were employed (653), and 1.2% were retired from service. However, no significant association was observed among genders and educational levels in regard to acceptance of the booster vaccine. The outcomes of the study revealed that the increased acceptance of booster doses of the SARS-CoV-2 vaccines among the public was associated with the intent of personal and family protection. Moreover, individuals with low socioeconomic status and pregnant females showed the least acceptance towards the vaccine inoculation. The study also revealed a decline trend of accepting SARS-CoV-2 vaccine among children.

## 1. Introduction

COVID-19, caused by coronavirus SARS-CoV-2, has claimed billions of lives worldwide and has pushed more than 100 million below the poverty line due to lockdown [1]. The World Health Organization (2022) reports that 566 million people have been affected by the COVID-19 virus, resulting in more than 6 million deaths worldwide. These are statistics of cases that have been reported; however, several cases and deaths were not reported during the pandemic due to fear among people [2]. The World Health Organization (WHO) declared the COVID-19 pandemic on 11 March 2020 [3].

COVID-19 virus is expected to impose a huge burden on the economy worldwide, and the only way to combat the virus is to develop herd immunity via the natural way or through immunization [4]. The release of the genetic content of the COVID-19 virus on 11 January 2020 resulted in the rapid development of the vaccine [5]. COVID-19 vaccines were rapidly prepared in a quiet little span of time due to the advancement of technology and research [6]. BNT162b2 was the first nanoparticle-formulated nucleoside-modified mRNA vaccine which received approval in December 2020 for emergency use in the United Kingdom, Bahrain, Canada, Mexico, Saudi Arabia, and the USA (Lamb 2021). Subsequently, several other vaccines were available for use, including Moderna, Cansino, AstraZeneca, Novavax, Shifa Pharmed, Sanofi Pasteur, and Sinopharm [7].

Twelve billion two hundred vaccine doses had been administered by June 2022 (WHO 2022), and people still hesitate to get vaccinated due to misconceptions. Several factors, including past experience, risk perception, education level, knowledge, and religious and moral values, lead to reluctance to get vaccinated [8]. The durability and efficacy of the vaccine were questionable, as it was developed in an emergency; secondly, vaccine approval was immediately followed by false claims and rumors spread through media that further hindered its acceptance among the public [9]. Some health workers also refused to accept vaccines initially, thus further increasing the doubts among the public [10]. These factors led to public hesitancy towards the vaccine [11]. Every other child in Sindh, Pakistan, missed his/her routine vaccine during the pandemic due to misconceptions surrounding the vaccine [12].

Khan, et al. [13] state that Pakistan is a country where it is challenging to convince people to get vaccinated. Khan, Mallhi, Alotaibi, Alzarea, Alanazi, Tanveer and Hashmi [13] further add that Pakistan has a weak healthcare system, dealing with a dense population, which leads to a faster spread of viral diseases. Thus, in the situation of the COVID-19 pandemic, convincing people to get vaccinated is the second major problem faced by the country besides the pandemic itself.

Initially, the people of Pakistan were reluctant towards the vaccine, but with time, misconceptions regarding the vaccine have been proved wrong by the fact that they have saved many lives. This has led to the acceptance of vaccines in Pakistan. Asian countries, including China, South Korea, India, and Singapore, were among those countries that showed the highest trust in government regarding the efficacy of the vaccine, with more than 80% of individuals willing to take the dose [14]. Moreover, 71% of the population in Pakistan willingly wants to get vaccinated against the deadly virus [15].

A vaccination program can only be successful if at least 60–70% population of a country is willing to get a dose, which will develop herd immunity [16]. Pakistan is a low-income country, with 35% of the population living below the poverty line, so in order to achieve vaccination goals, the government facilitated free vaccinations for all [17]. The majority of the Pakistani population preferred the Sino Pharm vaccine that was prepared by a pharmaceutical company in China, with around 48% of people waiting for their free vaccination dose, whereas about 52.7% were willing to pay less than 500 PKR for a COVID-19 vaccination [17].

According to Covid.gov.pk (accessed on 13 January 2023), more than137 million people have received just their first dose, and about 130.62 million people have been fully vaccinated in Pakistan. Among these individuals, only 48.04 million individuals took booster doses [18]. This indicates that people are reluctant towards taking booster doses, as it is believed that the initial two doses are enough to boost natural immunity among an individuals and can protect them against the disease [19]. A limited effort has been put towards booster doses in Pakistan. Even in high-socioeconomic countries such as China and the United States, acceptance of booster doses is low compared to initial doses [20].

A recent study at Aga Khan University, Pakistan, concluded that the acceptance rate for the booster COVID-19 vaccine is 67% among people in Pakistan [21]. Different factors control the perception of the people. Vaccination was also declared compulsory in schools in many countries, including Pakistan [22]. Moreover, healthcare professionals who were in direct contact with COVID-19 patients preferred getting vaccinated [5].

The administration of a third booster dose is being enforced due to the rising risk of the B.1.617.2 delta variant of the COVID-19 virus [23]. A survey showed that 66% of people in Pakistan were willing to pay for a booster dose, whereas, in China, about 70% of individuals were not willing to pay for the booster dose and preferred a free dose [24]. Acceptance of the booster dose was also found to be higher in Pakistan (89.4%) compared to China (84.8%) and America (79.1%) [24]. Seboka, et al. [25] study also indicates that people in Pakistan are now 78% less hesitant towards getting boosters compared to in the early days of the pandemic. A booster dose was accepted regardless of sex, age, and educational level, with more acceptances among people in the age group 18–30 (80%). Only individuals who were under the age of 18 (72.5%) were hesitant towards the booster dose. Consistent with the study, women were reluctant towards the booster doe due to social media reporting chances of damage to reproductive organs, even though very little data are available related to this concept [26].

This study aimed to collect data regarding the perception of people towards COVID-19 vaccinations and to assess their attitude and preference regarding booster doses of the virus.

## 2. Materials and Methods

This study was carried out to evaluate the perception, knowledge, and acceptance of the general public towards the COVID-19 vaccine, in general, and booster dose of COVID-19, in particular. To evaluate this, a questionnaire was designed and sent out to some of the faculty members of Riphah International University and University of Veterinary and Animal Sciences (UVAS) for initial assessment. The suggestions of the members were reviewed, and the questionnaire was modified accordingly. The finalized questionnaire was sent out online, through Google forms. Some of the questionnaires were filled face-to-face at different hospitals. The ethical approval of the study was taken from Institutional Review Board of Riphah International University, Lahore, Pakistan (RCVetS-1090). Figure 1 reveals the inclusion and exclusion criteria of the individuals in the study. Moreover, STROBE checklist was designed to improve the quality and transparency of the cross-sectional study (Appendix A). 

The questionnaire remained open for response from April 2022 to July 2022. After that, the form response was closed, and the results were exported and compiled in a CSV file format. The results were evaluated with SPSS v21 and MedCalc v20 statistical software. Any incomplete forms were excluded from the study. Moreover, the participants who did not have a COVID-19 vaccine administered were also excluded from the main study; however, we kept their initial responses in order to evaluate the other factors. The results were displayed in medians, percentages (%) expressed in 95% confidence intervals (95% CI). The detailed statistical analysis was performed by odds ratios, adjusted odds ratios, and a Chi-squared test where appropriate. Results were considered statistically significant if the *p*-value was less than 0.05.

## 3. Results

A total of 982 participants’ responses were received (Table 1). The responses of 3 participants were removed since they were incomplete, so only 979 respondents’ data were evaluated. The median age of the respondents was 22 (95% CI: 18 to 35) years, with age ranges between 16 and 74 years. In total, 409 (41.8%) females and 570 (58.2) males were part of the study. The highest number of the participants was from the province of Punjab (84.5%), followed by Islamabad (3.8%), Sindh (3.7%), Khyber Pakhtunkhwa (2.7%), Baluchistan (2.6%), Gilgit Baltistan (1.4%), and Azad Jammu and Kashmir (1.4%).

On the basis of formal education, the highest number of respondents had at least a bachelor’s degree (54.4%), followed by postgraduate’s degree (master’s = 17.9%; PhDs or above = 5.4%), intermediate level degree (intermediate = 16.4%; diploma = 2.2%), matriculate (2.2%), and below matriculate (1.3%). Moreover, 27.3% of the participants were employed or engaged in some sort of business, 71.6% were unemployed, and 1.1% were retired. Among the employed persons, 20.3% have a job which does not involve public interaction, whereas the rest of the 79.7% individuals have an interactive profession. Moreover, 39.9% of the participants are from healthcare or allied healthcare professions, whereas the rest are from non-health professions. Upon questioning about the status of COVID-19 vaccine administration, 93.5% of the total participants reported having had a vaccine shot, whereas 3.9% did not have a vaccine shot. In total, 2.7% of the participants were unsure whether they have been administered a vaccine or not. Based on this question, we segregated the participants, and those who were not sure or who did not have a shot of the vaccine were not asked any other questions and were excluded from the rest of the questionnaire. However, we kept them in the overall study to try to relate their demographics to vaccine acceptance.

It was observed that more subjects with interactive professions (78.6%) had received a vaccine than those in the non-interactive professions (15.8%) (*p* < 0.05).

The results were then recalculated after removing the individuals with no vaccine administered or who did not remember whether they were administered a COVID-19 vaccine. The mean age of the COVID-19 vaccine recipients was 24.49 ± 0.24 years (range: 16.0–65.0 years), and 41.5% of these were female, whereas 58.5% were males. However, no significant difference was observed between gender and the acceptance of the vaccine.

Similar to previous trends, the highest number of the participants who were administered the vaccine were from the province of Punjab (85.8%), followed by Islamabad (3.9%), Sindh (2.8%), Khyber Pakhtunkhwa (2.6%), Baluchistan (2.3%), Gilgit-Baltistan (1.3%), and Azad Jammu and Kashmir (1.2%).

The educational profile also remained the same, with the highest number of vaccine recipients having at least a bachelor’s degree (56.0%), followed by postgraduates (master’s = 18.6%; PhDs or above = 5.4%), intermediate level degree (inter = 15.8%; diploma = 1.9%), matriculate (1.3%), and below matriculate (1.1%). Furthermore, differences among the responses of the doctorate-level individuals are represented in Table 2.

Statistical results of doctorate (Ph.D.) participants related to the acceptance of COVID-19 vaccines are available as Appendix A. 

A total of 251 (27.4%) participants were employed, 653 (71.4%) were unemployed, and 11 (1.2%) were retired. Among the employed persons, 42 (16.7%) had a job which did not involve public interaction, whereas the rest of them, i.e., 209 (83.3%) individuals, had a socially interactive profession. In total, 83 (39.7%) of the participants were from healthcare or allied healthcare professions, whereas the rest were from non-health profession (60.3%).

A total of 915 participants had a COVID-19 vaccine administered: 12.3% had only one dose of the vaccine; 62.2% of the participants had one booster shot, along with the first dose; and 25.5% had a first dose and two booster doses administered (Figure 2). Females and healthcare professionals showed higher acceptance of the booster dose (ORs: 1.167 and 1.518, respectively). Interestingly, individuals without an interactive profession had higher acceptance (OR: 19.90) of booster doses. Similarly, the Chi-square test revealed significant differences of proportions among the vaccine acceptance among individuals of various education levels (*p* < 0.0001) and different provinces (*p* < 0.0001). Moreover, no significant difference was observed among individuals in relation to employment status (0.6037). Nearly 71.1% of the participants were sure that they never had an infection of COVID-19, 18.1% had had a prior infection, and 10.7% were not sure whether they have ever contracted the infection.

When asked about prior medical issues, 59.7% reported morbidities, and 40.3% mentioned that they are completely healthy.

Among the patients with prior illnesses, 127 (13.9%) had asthma, 87 (9.5%) had immune-related disorders, 86 (9.4%) had hypertension, 77 (8.4%) had a neurological disorder, 74 (8.1%) had obesity, 31 (3.4%) had diabetes, 28 (3.1%) had other cardiovascular disorders, 27 (3.0%) had COPD, 20 (2.2%) females were diagnosed with pregnancy, 19 (2.1%) had hepatic disorder, and 14 (1.5%) were diagnosed with chronic kidney disorders.

In all, 82 (9%) of the total respondents reported that they had observed some side effects after the third dose of the vaccine. However, when these participants were questioned about which of the doses they experienced more side effects from, most of the respondents (40.2%) reported that they felt most unwell after the second dose of the vaccine, followed by the third dose (35.4%), whereas the first dose had the least number of side effects reported (24.4%).

When questioned regarding the acceptance of the third booster dose of the vaccine, 280 (30.6%) respondents displayed their interest in getting a vaccine booster, 216 (23.6%) were not sure whether they would go for a booster, 197 (21.5%) did not want a booster dose, and only 131 (14.3%) had already had a booster dose received. Moreover, 91 (9.9%) respondents reported that they will only go for a booster dose if it is required by their employer.

Those who had already taken a third booster dose were asked for their reason for taking the booster. In all, 72 (55.0%) respondents reported that they received the vaccine booster for their own protection, 34 (26.0%) mentioned that they received it to ensure the well-being of their family, 15 (11.5%) had it because of travel requirement, and the rest of the respondents had it because their employer required they receive it.

No significant relationship was observed when we compared the acceptance of the booster vaccine and the educational level of the participants. 

For all the participants who had a shot of the COVID-19 vaccine, we inquired about what incentivized them to get the vaccine. A total of 524 (57.3%) reported that the offer of a free vaccine from the government encouraged them to get the vaccine, while the rest of them, i.e., 391 (42.7%), had it because of the compulsions from their employers or travel requirements. Furthermore, most of the participants (50.1%) mentioned that they were self-motivated for the vaccine administration, 278 (30.4%) responded that they were motivated by their family members, 89 (9.7%) were motivated by their employers, and 5.2% were motivated by their friends; caregivers were also among the individuals who encouraged the subjects to get vaccinated.

Upon being asked whether the booster dose would be beneficial or not, 465 (50.8%) thought that it is beneficial, 334 (36.5%) participants were unsure, and 116 (12.7%) participants thought that the booster dose of the vaccine is not beneficial.

A similar question regarding the efficacy of vaccine was asked from the interviewees. A similar trend was observed in response, with 553 (60.4%) who thought that the vaccine was efficacious and 104 (11.4%) who thought otherwise; meanwhile, the rest (28.2%) were unsure about the efficacy.

Moreover, we asked the participants what their source of information was for assessing the effectiveness of the vaccine. In all, 265 (29.0%) mentioned that their source of information was the information disseminated by the health department, 101 (11.0%) participants reported that they heard it from their local community, and 99 (10.8%) reported their caregivers as their source of information. Social media (55.2%), television (21.3%), and newspapers (8.3%) were also a source of information for the assessment of vaccine efficacy (Table 3).

Furthermore, we asked the participants which of the vaccines they had been administered. In all, 22 (2.4%) had AstraZeneca, 162 (17.7%) had Pfizer, 62 (6.8%) had CanSino, 29 (3.2%) had CoronaVac, 57 (6.2%) had Moderna, 425 (46.4%) had SinoVac, 206 (22.5%) had SinoPharm, and only 9 (1.0%) had Sputnik administered. The rest of the participants (0.9%) were unsure of the vaccine they had been given.

Upon being questioned as to whether they would recommend others to get a booster vaccine, 555 (60.7%) of respondents reported that they would recommend others to get completely vaccinated, 289 (31.6%) responded that one must only go for a booster dose if needed, and 71 (7.8%) mentioned that they would not recommend others to get a vaccination booster.

The next question asked was their opinion about the booster vaccine for children. Most participants (39.8%) responded that natural immunity is the best way for children to achieve immunity against COVID-19 infection rather than having a vaccine. In total, 339 (37.0%) thought that children must not be vaccinated for COVID-19, whereas 212 (23.2%) respondents believed that children must be vaccinated against the COVID-19.

The same question was asked in regard to students and whether they should be compelled to get the vaccination. Mostly, people (59.8%) believed that it must be mandatory for students to get a vaccination, and 246 (26.9%) believed that it is an independent choice of an individual and that ethical approval must be there for vaccine administration. On the contrary, 13.3% believed that there must be no compulsion for students regarding vaccination.

When asked whether the vaccine may be given during pregnancy, a large number of respondents (43.9%) said that it must not be given, 317 (34.6%) patients were not sure, and only 196 (21.4%) of the participants thought that the vaccination is safe and may be given to females during pregnancy.

The respondents were asked about their view on compulsory vaccination. Most of the participants (71.2%) of the study thought that it was a good decision to have compulsory vaccinations. Half of the remaining subjects said that it was hard for them to agree with mandatory-vaccination views, and the other half believed that the decision of compulsory vaccine is not an ideal approach and that people must have freedom to choose what they want to do.

Surprisingly, 94 (10.3%) individuals claimed that COVID-19 vaccine administration had resulted in sleep deprivation for them, while 179 (19.6%) reported that, for a few weeks, they had faced sleep-cycle disturbances. However, for most of the participants (70.2%), there was no effect on sleep after vaccine administration. Similarly, a few individuals (9.2%) reported that the vaccine had also altered their mental health. However, most of the participants (80.0%) did not have any effect or are not sure if their mental health is affected by the COVID-19 vaccine.

## 4. Discussion

The COVID-19 vaccines have fought significantly against the disease and have been proven to be safe and effective throughout the world. There have been huge campaigns and efforts by all the governments of the world to spread this information and increase vaccine acceptance among people, yet a significant number of people show unwillingness towards accepting the vaccine due to certain barriers. To effectively reduce the transmission of the virus, herd immunity needs to be created, and the required vaccination rate to achieve this target is 82% [27]. This requires the identification of the factors causing the vaccine hesitancy on an individual and population level and calls for policymakers to develop appropriate guidelines and conceptual frameworks. Based on those tailored policies, the public health authorities will be able to raise awareness campaigns targeting the vaccine hesitancy and improving the rate of vaccine acceptance [28]. This study was carried out to evaluate the perception, knowledge, and acceptance of the general public towards the COVID-19 vaccine, particularly the third booster dose, and looked at a total of 979 respondents residing all over Pakistan, including Azad Jammu and Kashmir.

We found that the highest number of respondents who showed vaccine acceptance belong to Punjab, as it is the most densely populated province of Pakistan. The majority of vaccine recipients held a bachelor’s degree, followed by a master’s degree. The people who had a history of COVID-19 infection and had received a vaccine against it or had a history of receiving any other type of vaccine showed more acceptance towards the booster dose of the COVID-19 vaccine. Studies conducted in Bangladesh presented similar findings [29,30], where people who received the influenza vaccine earlier did not hesitate in regard to receiving the COVID-19 vaccine.

The side effects from the vaccine were reported by some recipients who participated in this study. Slightly more than half of the respondents reported comorbidities, but none of them suffered from critical or life-threatening conditions. There is evidence that side effects can be escalated in the case of comorbidities; however, the side effects did not last long and were usually reversed in 2–7 days. A study conducted on the side effects of the vaccine booster dose in Vietnam reported that the adverse effects of booster does were similar to those reported for the first and second doses of the vaccine. The study also reported that recipients of different vaccines, i.e., Moderna and Pfizer, reported different side effects [31]. Another study in Algeria presented results which are in line with the findings of our studies that minor adverse effects such as headache, nausea, fever, and body pains were experienced by the recipients for a short time [32].

The people in more interactive professions and those with a history of COVID-19 infection had a much higher rate of vaccine acceptance than those in the non-interactive professions. This finding implies that social interaction has a positive impact on people’s vaccine acceptance. The acceptance of vaccine varies in different countries due to different sociocultural structures and philosophical and religious beliefs. A survey of the global acceptance patterns of the COVID-19 vaccines found out that, in Asian (south and southeast) countries, 80% of people had trust in the vaccine as compared to those in European countries [14].

According to the findings of our study, the majority, i.e., 53.9%, of participants responded that they got vaccinated in order to protection their own health, followed by those who did so to protect the health of their family and those who did so to meed job requirements. The inclination towards receiving the vaccination is greater in healthcare workers as compared to others. The awareness in people with a non-healthcare background is relatively low, and they are obliged by personal or social reasons to take the vaccine. A survey conducted in the Southeast Asia and South Asian region revealed that more than 95% of healthcare workers showed willingness to receive vaccine against COVID-19 [33]. A review of the acceptance of the COVID-19 vaccine in Chinese healthcare workers predicted a high acceptance rate among them upon the availability of the vaccine [34]. Similarly, the findings of Trabucco Aurilio, et al. [35] stated that acceptance was significantly related to people’s jobs, with physicians showing the highest rate of uptake compared to other occupations. Moreover, vaccine hesitancy was positively associated with the fear of vaccination side effects and negatively related to confidence in the efficacy and safety of the vaccine [35].

To increase the vaccine acceptance among people, the right advertisement at the right time plays the role of a cornerstone. The majority of respondents in our study received information about vaccination efficacy through digital media, i.e., television and social media. Moreover, 60% of the participants further added that they will recommend getting the vaccine to others. In terms of designing and setting up interventions to enhance vaccine acceptance, healthcare professionals and their representative groups should be closely engaged in policymaker and health authority decisions regarding the establishment and implementation of vaccine recommendations [36].

However, the response was relatively low for the children and pregnant females. The low vaccine acceptance towards a vulnerable population such as children and pregnant females could be due to the side effects that may be aggravated in them. In a survey, only 37% of pregnant females showed acceptance towards receiving the vaccine if recommended due to potential harm to the fetus [37]. The safety of booster doses in children of 5–11 years has been studied, and it has been reported as safe when received by children; however, hesitancy was found most among people with low socioeconomic status [38].

Nevertheless, one of the best ways, as suggested by other researchers [39,40] as well, is to achieve herd immunity, and for that, nearly 80% of the overall population of Pakistan must be protected through the vaccination.

## 5. Conclusions

Overall, the participants of this study showed acceptance towards the third (booster) dose of the COVID-19 vaccines and a positive inclination towards promoting the cause for the protection of their friends and family. The participants with low socioeconomic status showed hesitancy towards receiving the vaccine, and the recommendation for their use in pregnant females was low. Overall, significantly higher proportions of respondents were reluctant to get their children vaccinated against COVID-19. The side effects from the booster dose were similar to the second dose but stayed for a shorter term and reversed quickly. To increase the vaccine acceptance, the mindset of the general population needs to be changed. It is suggested that the government should run campaigns to raise vaccine awareness, strategically targeting the people in low-socioeconomic areas, as a majority of them are hesitant. The awareness should also be created through digital media from trusted sources, such as the department of health, working with the local bodies in the targeted regions.

## 6. Limitations

This study is a cross-sectional survey which lacks a connection with the retrospective or real-time data. Moreover, the online questionnaire may have excluded those with limited internet access. The responses were supplied only by literate members of the society, and those with a disability or social disadvantage were not included in the study.

## Figures and Tables

**Figure 1 vaccines-11-01238-f001:**
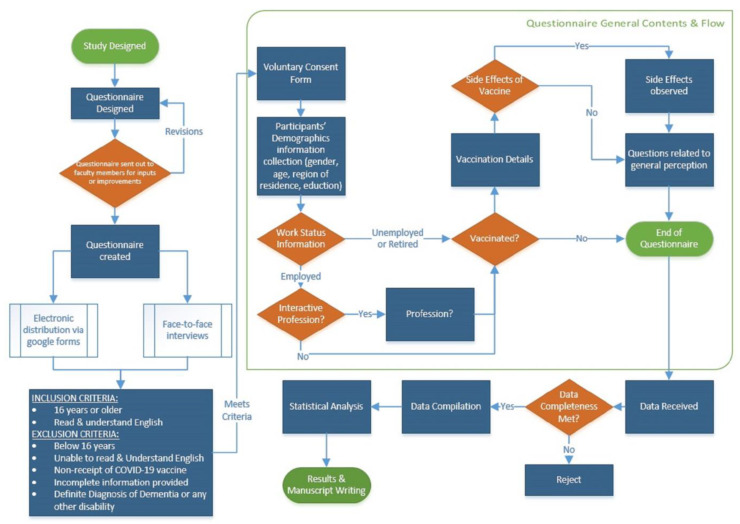
The general structure and workflow diagram of the study. The data regarding acceptance of SARS-CoV-2 vaccine were recorded through a questionnaire-based approach. The data were strictly observed to meet the inclusion and exclusion criteria prior subjected to statistical analysis (Appendix A).

**Figure 2 vaccines-11-01238-f002:**
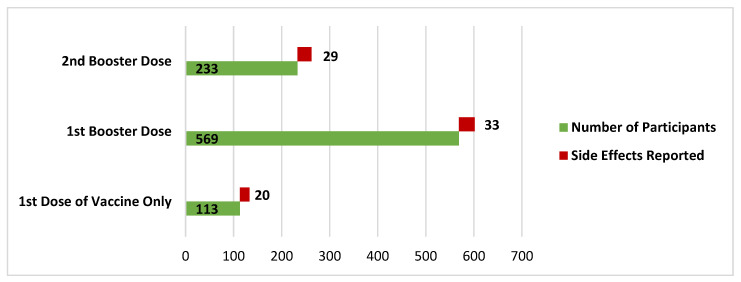
The differences among the proportions of COVID-19 vaccination doses and side effects.

**Table 1 vaccines-11-01238-t001:** The data of responses from the individuals included in the study regarding various associated factors.

Parameters		Never Received Vaccine [Row Wise Percentage] (95% Confidence Interval)	Vaccine Received [Row Wise Percentage] (95% Confidence Interval)	Total [N = 979] (%)	Adjusted Odds Ratio (95% CI) *p*-Value	Crude Odds Ratio and Chi-Square Results
Age	(Median ± 95% C.I)	22 (21.19 to 23.0) y	22 (22.000 to 23.0) y	22 y	-	-
Gender	Female	7.1% (10–4.99)	92.91% (95.02–90.01)	409 (41.8%)	0.86 (0.52–1.43) *p*-value 0.553	Odds ratio: 1.167 (95% CI: 0.7135 to 1.934) *p*-value 0.5534
	Male	6.15% (8.42–4.45)	93.86% (95.56–91.59)	570 (58.2%)	1.17 (0.7–1.94) *p*-value 0.553	
Province	Azad, Jammu, and Kashmir	21.43% (47.59–7.58)	78.58% (92.43–52.42)	14 (1.4%)	0.61 (0.09–4.35) *p*-value 2.28	Chi-square test *p*-value < 0.0001
	Baluchistan	16% (34.66–6.41)	84% (93.6–65.35)	25 (2.6%)	1.43 (0.27–7.57) *p*-value 2.28	
	Gilgit-Baltistan	14.29% (39.95–2.54)	85.72% (97.47–60.06)	14 (1.4%)	1.63 (0.23–11.7) *p*-value 1.68	
	Islamabad	2.71% (13.83–0.14)	97.3% (99.87–86.18)	37 (3.8%)	9.78 (0.92–104.58) *p*-value 1.39	
	Khyber Pakhtunkhwa	7.7% (24.15–1.37)	92.31% (98.64–75.86)	26 (2.7%)	3.27 (0.48–22.47) *p*-value 1.73	
	Punjab	5.08% (6.8–3.78)	94.93% (96.23–93.21)	827 (84.5%)	5.1 (1.37–18.97) *p*-value 5.46	
	Sindh	27.78% (44–15.85)	72.23% (84.16–56.01)	36 (3.7%)	0.71 (0.16–3.08) *p*-value 3.55	
Education	Bachelors	3.94% (5.95–2.6)	96.07% (97.41–94.06)	533 (54.4%)	7.32 (1.87–28.56) *p*-value 0.004	Chi-square test *p*-value < 0.0001
	Diploma	22.73% (43.44–10.13)	77.28% (89.88–56.57)	22 (2.2%)	1.02 (0.2–5.21) *p*-value 0.981	
	Doctorate	7.55% (17.86–2.98)	92.46% (97.03–82.15)	53 (5.4%)	3.68 (0.71–19.03) *p*-value 0.121	
	Intermediate	9.94% (15.54–6.22)	90.07% (93.79–84.47)	161 (16.4%)	2.72 (0.68–10.91) *p*-value 0.158	
	Masters	2.86% (6.52–1.23)	97.15% (98.78–93.49)	175 (17.9%)	10.2 (2.13–48.86) *p*-value 0.004	
	Matriculation	45.46% (65.35–26.93)	54.55% (73.08–34.66)	22 (2.2%)	0.36 (0.08–1.68) *p*-value 0.193	
	Below Matric	23.08% (50.26–8.18)	76.93% (91.83–49.75)	13 (1.3%)	0.27 (0.05–1.41) *p*-value 0.121	
Work status	Employed	6% (9.52–3.73)	94.01% (96.28–90.49)	267 (27.3%)	0.87 (0.4–1.5) *p*-value 0.633	Chi-square test *p*-value 0.6037
	Retired	0% (25.89–0)	100% (100–74.12)	11 (1.1%)	0.87 (0.48–1.56) *p*-value 0.633	
	Unemployed	6.85% (8.97–5.21)	93.16% (94.8–91.04)	701 (71.6%)	1.15 (0.64–2.07) *p*-value 0.633	
Interactive profession	No	22.23% (34.94–13.2)	77.78% (86.81–65.07)	54 (5.5%)	19.91 (5.38–73.63) *p*-value 0	Odds ratio: 19.90 (95% CI: 5.869 to 67.32) *p*-value < 0.0001
	Yes	1.42% (4.08–0.39)	98.59% (99.62–95.93)	212 (21.7%)	0.05 (0.01–0.19) *p*-value 0	
Professions	Health or allied healthcare	2.36% (8.18–0.42)	97.65% (99.59–91.83)	85 (8.7%)	0.66 (0.09–4.76) *p*-value 0.679	Odds ratio: 0.6587 (95% CI: 0.091 to 4.76) *p*-value 0.6794
	Others	1.57% (5.52–0.28)	98.44% (99.73–94.49)	128 (13.1%)	1.52 (0.21–11.02) *p*-value 0.679	

**Table 2 vaccines-11-01238-t002:** The data of the doctorate-level individuals regarding acceptance towards COVID-19 vaccination.

Acceptance of Doctorate Level Individuals towards COVID-19 Vaccines		
Responses	Percentage (%)	95% CI
Already vaccinated	30.19%	19.52–43.54
Maybe	11.33%	5.3–22.58
No	24.53%	14.94–37.57
Only if job requirement	7.55%	2.98–17.86
Yes	18.87%	10.59–31.36
Response missing	7.55%	2.98–17.86
Total	*n* = 53	

**Table 3 vaccines-11-01238-t003:** Percentage distribution of participants’ responses to the questionnaire regarding various aspects of SARS-CoV-2 vaccine acceptance and general attitude of the recipients.

Observed Factor	N (%)	95% CI
What is your reason for taking a third (booster) dose?		
Job requirement	100 (10.9%)	9.82–14.18%
Protection of colleagues	16 (1.7%)	1.17–3.05%
Protection of family health	167 (18.3%)	17.2–22.56%
Protection of my own health	493 (53.9%)	54.93–61.56%
Travel requirement	66 (7.2%)	6.18–9.81%
All of above	4 (0.4%)	0.19–1.21%
Which of the following would encourage you to get the vaccination?		
Free vaccination	524 (57.3%)	54.04–60.44%
Only if compulsory requirement for work or travel	391 (42.7%)	39.57–45.97%
Do you think that the booster is beneficial against severe infection?		
May be	334 (36.5%)	33.45–39.68%
No	116 (12.7%)	10.68–15%
Yes	465 (50.8%)	47.59–54.05%
Have you found the vaccine to be effective against the virus?		
Not Sure	334 (36.5%)	33.45–39.68%
No	116 (12.7%)	10.68–15%
Yes	465 (50.8%)	47.59–54.05%
What was your source of information for assessing the effectiveness of the vaccine?		
Newspapers	76 (8.3%)	4.78–7.38%
Social Media	505 (55.2%)	36.85–42.2%
Television	195 (21.3%)	13.39–17.33%
Employer	38 (4.2%)	2.18–4.06%
Healthcare system advertisements	265 (29.0%)	18.59–23.03%
Caregiver	99 (10.8%)	6.4–9.34%
Local community	101 (11.0%)	6.55–9.51%
Who motivated you to get vaccinated?		
Family	278 (30.4%)	27.49–33.44%
Friends	48 (5.2%)	3.98–6.89%
Job requirement	89 (9.7%)	7.98–11.82%
My Care Giver	42 (4.6%)	3.42–6.15%
Myself	458 (50.1%)	46.83–53.29%

## Data Availability

All the data are available.

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
