# Peer review of "Attitude and Acceptance towards COVID-19 Booster Doses among Literacy Advantaged Population in Pakistan: A Cross-Sectional Study"

_vaccines, 2023, doi:10.3390/vaccines11071238_

Round 1

Reviewer 1 Report

In this study, they to evaluate the perception, knowledge, and acceptance of general public towards the COVID-19 vaccine, in general, and booster dose of COVID-19. The study is interesting, but it has many limitations, more analysis is needed, and the methods section is incomplete. Finally, the results are not representative of the population.

Major commets: 

The introduction is too long.

Missing to include the objective of the study.

Methods

This section is incomplete. Include the STROBE check list (as an Appendix), and verify if they are complying with all the items on the STROBE check list.

Include study design.

What were the inclusion and exclusion criteria?

What was the result of the evaluation of the questionnaire of the 136 teachers?

Explain if the questionnaire was pilot tested.

Have the participants completed an informed consent?

It remains to include the study population, the sample and the sampling.

Statistic analysis.

Have confidence intervals (CI 95%) been used?

What was the estimated p-value?

Why did you use Chi-square? What were the comparisons you made? The most appropriate would be to use crude OR and adjusted OR, instead of Chi-Square.

Results 

The results are poorly described

Paragraphs 1, 2, 4, 5 and 6 are repetitive of Table 1.

Vaccination rates should be estimated with 95% CI.

Factors associated with vaccination should be compared between groups. This comparison must be made with crude OR and Adjusted OR.

The other paragraphs are also repetitive of Table 2 (it does not have a number). 

Authors need to perform further analysis.

Discussion

Authors must include a paragraph describing the public health implications.

Conclusions.

The data is not representative.

The authors presented conclusions on the basis of descriptive analyses. The authors should improve these conclusions on the basis of more robust analyses.

Limitations. Authors should describe any limitations of their study. Furthermore these limitations should be discussed.

Author Response

In this study, they to evaluate the perception, knowledge, and acceptance of general public towards the COVID-19 vaccine, in general, and booster dose of COVID-19. The study is interesting, but it has many limitations, more analysis is needed, and the methods section is incomplete. Finally, the results are not representative of the population.

Major comments:

Comment 1: The introduction is too long. Missing to include the objective of the study.

Reply: The authors are grateful to the reviewer for their suggestion. We have summarized the introduction and added the objective of the study in the last paragraph of introduction.

Methods

Comment 2: This section is incomplete. Include the STROBE check list (as an Appendix), and verify if they are complying with all the items on the STROBE check list.

Reply: Strobe checklist has been added as Appendix and verified for compliance with all the items on the checklist. 

Comment 3: Include study design.

Reply: We thank the reviewer for pointing us out this issue. The study design has been updated in the article as figure 1.

Comment 4: What were the inclusion and exclusion criteria?

Reply: Figure 1 has been updated to describe the inclusion and exclusion criteria. Moreover, line # 128 to line# 130 also describes the criteria in detail.

Comment 5: What was the result of the evaluation of the questionnaire of the 136 teachers?

Reply: Evaluation of questionnaire of the teachers and doctoral level subjects has been updated in tabular form in the Table 2.

Comment 6: Explain if the questionnaire was pilot tested.

Reply: Questionnaire was pilot tested, updated through repeated suggestion and results.

Comment 7: Have the participants completed an informed consent? It remains to include the study population, the sample and the sampling.

Reply: Yes, the participants have filled an informed consent.  

Statistical analysis:

Comment 8: Have confidence intervals (CI 95%) been used?

Reply: The statistical analysis has been updated with 95% CI.

Comment 9: What was the estimated p-value?

Reply: The p-values of statistical tests have been updated in the results.

Comment 10: Why did you use Chi-square? What were the comparisons you made? The most appropriate would be to use crude OR and adjusted OR, instead of Chi-Square.

Reply: The authors are highly thankful to the reviewer for their valuable suggestion. The methodology and results sections have been updated with OR. Moreover, Chi-square test where applicable was used to demonstrate the differences among the proportions of various study groups.

Results:

Comment 11: The results are poorly described.

Reply: Thank you for the comment. The results have been updated accordingly.

Comment 12: Paragraphs 1, 2, 4, 5 and 6 are repetitive of Table 1.

Reply: The table has been elaborated in the paragraphs to describe the factor wise results of the study.

Comment 13: Vaccination rates should be estimated with 95% CI.

Reply: Vaccination rates have been estimated with 95% CI.

Comment 14: Factors associated with vaccination should be compared between groups. This comparison must be made with crude OR and Adjusted OR.

Reply: Factors associated with vaccination have been compared between groups by using the Odd ratio.

Comment 15: The other paragraphs are also repetitive of Table 2 (it does not have a number). 

Reply: The tables have been numbered and the paragraphs are further elaboration of the tabular results.

Comment 16: Authors need to perform further analysis.

Reply: Further statistical analysis has been performed on the data according to the valuable suggestion of the reviewer.  

Discussion:

Comment 17: Authors must include a paragraph describing the public health implications.

Reply: line # 346 to line # 349 have been added to the discussion section describing the public health implications.

Conclusions:

Comment 18: The data is not representative. The authors presented conclusions on the basis of descriptive analyses. The authors should improve these conclusions on the basis of more robust analyses.

Reply: Data analysis has been subjected to the odd-ratios regarding the associated factors and the data regarding vaccination has been updated in the form of 95% CI. Conclusions have been updated based on the results of data analysis.

Limitations: Authors should describe any limitations of their study. Furthermore, these limitations should be discussed.

Reply: Limitations have been discussed under the “Limitation” section.

Reviewer 2 Report

The submitted manuscript presents “Attitude and acceptance towards COVID-19 booster doses among general public in Pakistan: a cross sectional study”. The authors conducted a questionnaire-based study, in which n=982 participants participated from all over Pakistan, to evaluate the perception knowledge, attitude, and acceptance of the local public towards the SARS-CoV-2 vaccine, in general, and a booster dose of SARS-CoV-2. The author found no significance among genders and educational levels with acceptance of the booster vaccine. This study will support to the national and international world to take immediate step to overcome the resurgence of COVID-19.

I found that non-educated/illiterate or population from remote areas did not participate in this study. This picture is not the mirror of whole population in Pakistan because of more participants were belonged to educator system either university or college. Please address this issue also.

I found few others minor typos and error in this manuscript. Please modify this accordingly as mentioned below.

The introduction is too general: Report the epidemiology of antimicrobial resistance that focus on the study area. Please show the study gaps.

Please also give a description of the statistical analysis (tool or other) used in this study. Also mention in table or text.

Line 79: United Kingdom (UK)

Please also provide the supplementary data.

Tables and Figures are without legends and caption. Please write this.

Please revise the conclusion and improve the discussion according to the results.

Author Response

The submitted manuscript presents “Attitude and acceptance towards COVID-19 booster doses among general public in Pakistan: a cross sectional study”. The authors conducted a questionnaire-based study, in which n=982 participants participated from all over Pakistan, to evaluate the perception knowledge, attitude, and acceptance of the local public towards the SARS-CoV-2 vaccine, in general, and a booster dose of SARS-CoV-2. The author found no significance among genders and educational levels with acceptance of the booster vaccine. This study will support to the national and international world to take immediate step to overcome the resurgence of COVID-19.

Comment 1: I found that non-educated/illiterate or population from remote areas did not participate in this study. This picture is not the mirror of whole population in Pakistan because of more participants were belonged to educator system either university or college. Please address this issue also.

Reply: The study was designed to aim the literate population who can read and understand English language. Therefore, it was not possible to include the responses from illiterate people. The most respondents had filled the questionnaire through online. This has been addressed as inclusion criteria in methodology. The present study (n= 982 participants) may reflect an overview of the attitude & behaviors of literate society in Pakistan. Currently this is the limitation of our study hence we have modified the title of study as it was misleading in this context.  

Comment 2: The introduction is too general: Report the epidemiology of antimicrobial resistance that focuses on the study area. Please show the study gaps.

Reply: Thanks for your valuable suggestion. The introduction section has been updated.

Comment 3: Please also give a description of the statistical analysis (tool or other) used in this study. Also mention in table or text.

Reply: Statistical analysis was performed using Odd ratios, Chi-square test and 95% CI of percentages. All analysis was done through SPSS and MedCalc windows software. Same has been mentioned in the text and tables.

Comment 4: Line 79: United Kingdom (UK)

Reply: The abbreviation has been updated.

Comment 5: Please also provide the supplementary data.

Reply: Supplementary data has been added.

Comment 6: Tables and Figures are without legends and caption. Please write this.

Reply: Legends and captions have been updated.

Comment 7: Please revise the conclusion and improve the discussion according to the results.

Reply: Thank you for pointing out this aspect. The authors have made the necessary changes as required by the reviewer.

Reviewer 3 Report

I'd like to thank for asking me to review this paper which was conducted to collect data regarding the perception of people toward COVID-19 vaccinations and to assess their attitude and preference regarding booster doses. Vaccine acceptance and hesitancy are relevant issue worldwide. The authors should read these published paper to deep discuss their findings and to make a comparison whith data from other countries:

doi: 10.3390/tropicalmed7120419

doi: 10.1016/j.ijnurstu.2022.104241

Author Response

Comment 1: The authors should read these published papers to deep discuss their findings and to make a comparison with data from other countries:

doi: 10.3390/tropicalmed7120419

doi: 10.1016/j.ijnurstu.2022.104241

Reply: The authors are highly thankful to the reviewer for their precious guidance. The authors have carefully read the said articles and have found these very helpful. Citations have been made in the discussion of this article. Introduction, Methodology & statistical analysis has been improved as per your valuable suggestions.

Round 2

Reviewer 1 Report

The authors have corrected the manuscript. However I have some concerns.

Details of the statistical analysis are lacking. They have not included the p-value in the statistical analysis section "Section Methods".

In the results section. Table 1. The p-values for the crude ORs have not been included. Adjusted ORs have not been calculated. For polytomous variables they must use a reference variable to calculate the OR.

Author Response

Comment 1. Details of the statistical analysis are lacking. They have not included the p-value in the statistical analysis section "Section Methods".

Reply. Thank you for highlighting this basic deficiency. The authors have updated the p-value in the “Methods Section”.

Comment 2. In the results section. Table 1. The p-values for the crude ORs have not been included. Adjusted ORs have not been calculated. For polychotomous variables they must use a reference variable to calculate the OR.

Reply: The authors are highly thankful to the reviewer for his kind guidance regarding the calculation of adjusted odd ratios related to the polychotomous variables using a reference variable. The authors have updated the p-values for crude odd ratios in the table 1. Moreover, the adjusted ORs have also been updated in the table as suggested by the reviewer.